# Model of Pathological Collagen Mineralization Based on Spine Ligament Calcification

**DOI:** 10.3390/ma13092130

**Published:** 2020-05-04

**Authors:** Sylwia Orzechowska, Renata Świsłocka, Włodzimierz Lewandowski

**Affiliations:** 1M. Smoluchowski Institute of Physics, Jagiellonian University, Łojasiewicza 11, 30-348 Kraków, Poland; 2Department of Chemistry, Biology and Biotechnology, Bialystok University of Technology, 15-351 Białystok, Poland; r.swislocka@pb.edu.pl (R.Ś.); w.lewandowski@pb.edu.pl (W.L.)

**Keywords:** calcification, collagen, computed microtomography, mineralization model, hydroxyapatite, ligamenta flava

## Abstract

The aim of the study was to determine the time of mineral growth in human spine ligaments using a mathematical model. The study was based on our previous research in which the physicochemical analysis and computed microtomography measurements of deposits in ligamenta flava were performed. Hydroxyapatite-like mineral (HAP) constituted the mineral phase in ligament samples, in two samples calcium pyrophosphate dehydrate (CPPD) was confirmed. The micro-damage of collagen fibrils in the soft tissue is the crystallization center. The growth of the mineral nucleus is a result of the calcium ions deposition on the nucleus surface. Considering the calcium ions, the main component of HAP, it is possible to describe the grain growth using a diffusion model. The model calculations showed that the growth time of CPPD grains was ca. a month to 6 years, and for HAP grains >4 years for the young and >5.5 years for the elderly patients. The growth time of minerals with a radius >400 μm was relatively short and impossible to identify by medical imaging techniques. The change of growth rate was the largest for HAP deposits. The mineral growth time can provide valuable information for understanding the calcification mechanism, may be helpful in future experiments, as well as useful in estimating the time of calcification appearance.

## 1. Introduction

Mineralization of soft tissues, such as aortic valve cusps, tendons and ligaments, is a progressive disorder which has been reported for many years [1,2]. It was considered as a passive process of calcium salts deposition in tissues. However, the use of advanced techniques enables a detailed analysis of mineralized areas of a tissue [3]. Recent studies have shown some similarities of pathological mineralization with actively controlled processes in bones [4,5,6,7]. Bone mineralization is the formation of hydroxyapatite on the biological matrix which is of collagen fibrils. Collagen fibrils occurring in bones demonstrate specific interfibrillar spaces (structural gaps) endowed with an electric charge where the nucleation of bone apatite can be started. Collagen is not a crystallization promoter but the process is assisted by matrix vesicles released from the osteoblast cell membrane [8,9]. The matrix vesicles are rich in proteins, phospholipids, calcium, and inorganic phosphates and provide the right concentration of Ca and P for crystal precipitation. During bone mineralization, collagen fibrils aggregate side by side in a lateral register, their gap zones also come to be in the register, and give rise to transverse, ‘electron-dense bands’ that cross the collagen bundles [9]. The effect of such fibril aggregation is the creation of electrically charged areas in which the hydroxyapatite crystals are formed. The collagen fibril aggregation manner reflects a crystal shape. It is known that after the start of nucleation in fibril gaps, crystal growth initially takes place within the structural gap and then covers the area outside the gap [10]. The similarity of the calcification mechanism in the presented model with physiological mineralization relies mainly on the presence of gap zones in collagen fibrils in which the nucleation of mineral grains occurs. The authors focused on the physical aspects of the ion deposition process as a consequence of the presence of structural micro-damages.

A significant role in pathological tissue calcification is attributed to mechanical stress (accumulation of microtrauma). The experimental studies performed on the spines of rats demonstrated that stretching and torsion movements result in the proliferation of cartilage tissue, transforming into bone tissue in the spinal ligaments [11,12]. Similarly, studies performed on patients exposed to stretching showed an increase in the rate of cytokines associated with the bone formation process (i.e., bone morphogenic protein, BMP) [11,12].

The influence of pathological spine curvatures, especially thoracic kyphosis, on the mineral deposits formed in spine ligaments has been confirmed. Abnormal spine curvature leads to changes in the strain distribution, especially in the part of the lower thoracic spine where the calcification of the ligaments is most frequent [13]. Structural micro-damage as a result of microtrauma of the tissue can be a place of mineral nucleation.

It has been suggested that ligament mineralization may be the cause of canal stenosis or vice versa—it is the result of degenerative changes [7,14]. Spinal stenosis is a condition in which there is a disproportion between the volume of the canal and the volume of the spinal cord, leading to compression of the spinal roots. As a result of the spinal canal narrowing, the spinal ligaments bend, which undergo hypertrophy and calcification [15]. Many authors have shown that mineralization of the spinal ligaments is correlated with spinal canal stenosis [7,16,17,18]. Histological results show that the degenerative changes in the spinal ligaments are manifested by a decrease in the amount of elastin and an increase in collagen content. This process is explained by defects and mechanical stress in the ligaments, resulting in further damage, and as a result elastin fragmentation and collagen proliferation [7].

Yellow ligaments (*ligamenta flava*, LF) are structures located between the arches of neighboring vertebrae. They close the spinal canal from the back (Figure 1). Due to their elasticity, they are of great importance for the statics and mechanics of the spine, stabilizing the spine in the sagittal plane and preventing excessive bending towards the front [19]. The non-typical structure of the yellow ligaments, consisting of a collagen to elastin ratio of 1:2, is a reason for a significant adaptation to mechanical loads, especially to stretching. This may explain the slight and incidental mineralization of LF. The prevention of soft tissue calcification is difficult because the presence of mineral deposits in the early stage of growth often does not give clinical symptoms. Described pathology is diagnosed in an advanced stage as accompanying other diseases. Therefore, the growth time of mineral deposits in soft tissues is difficult to estimate. The mineralization of the ligamenta flava in an advanced stage can cause neurological disturbances since the mineralized ligament can compress the spinal cord. The most typical symptoms are pain, reduced muscle strength with paresis, disturbances of feeling in the perianal region, as well as loss of control over micturition and defecation [20].

The mechanism of mineralization in tissues is described in the literature using several models [21,22,23,24,25,26,27]. However, the time of mineral growth in human soft tissues has not been discussed. According to Urry’s charge neutralization theory of calcification, calcium is bound to neutral sites of protein and phosphate while carbonate ion binding follows to produce charge balance. A circular dichroism pattern of solutions of organic acids demonstrates that calcium ions reverse conformational changes in the molecular structure. Neutral bonding sites on uncharged polypeptides bind cations by coordination with acyl oxygens and change the electrochemical properties of collagen. Binding of Ca ions is a chronic progressive process with age [28,29].

The aim of the study was to determine the time of mineral growth based on a diffusion model. The knowledge about the growth time of minerals in the yellow ligaments as well as in the case of other soft tissues has fundamental importance, giving information on the calcification mechanism. It supports the estimation of a relation between incidents in the human body (i.e., metabolism disease, stenosis of tissues, change of diet, and mechanical trauma related to sport), as well as the appearance of calcifications. Thanks to this knowledge the elimination or reduction of harmful factors is feasible. It should be stressed that the important advantage of the demonstrated model is the calculation based on the analysis of human samples using our experimental data [30] concerning the mineralization in human ligamenta flava. The models generally presented in the literature are based mainly on the analysis of laboratory animal subjects (rats, mice) which are characterized by faster metabolism than the metabolism in the human case. The use of large animals as models of calcification has still been discussed [21,31,32,33].

## 2. Materials and Methods

The 24 ligamentum flavum samples (LF), surgically extracted from the lumbar spine at the level of L1–L5 vertebrae, were analyzed. The biological material consisted of 15 samples of ligaments obtained from patients, aged (55–81) years old, qualifying for surgery due to spinal canal stenosis. The control group contained 9 samples of ligaments taken from patients aged 20–53 years old who had undergone mechanical spinal trauma. The physicochemical analysis of mineral deposits in LF was performed in our previous work [30] with the use of Fourier transform infrared spectroscopy (FTIR) and X-ray fluorescence (XRF) techniques. Moreover, the volume and density of mineral grains were calculated using computed microtomography (µ-CT) [30]. The obtained data were put to use in model calculations. The reconstructed µ-CT image of the ligamentum flavum with mineral deposits is shown is in Figure 2. The analysis was carried out for two chemical structures of deposits: hydroxyapatite-like minerals (HAP) and calcium pyrophosphate dehydrate (CPPD) in ligaments of elderly patients suffering from canal spinal stenosis and the young patients from the control group.

### 2.1. Assumptions and Calculations

The micro-damage of the collagen fibril is the center of the mineral grain crystallization. The growth of the mineral nucleus, located in the extracellular fluid environment, occurs as a result of calcium ions deposition on the nucleus surface. The elemental analysis of deposits performed in our previous work [30] confirmed that Ca was present in the highest concentration in the minerals.

Considering the calcium ions, which are the main component of the mineral phase in deposits, it is possible to describe the process of grain growth using a diffusion model. The proposed model will enable us to estimate the growth time of the mineral grains. The following assumptions were made:
A spherical symmetry of the system,The transport of calcium ions is the limiting factor for grain growth,No chemical reactions in the environment

Due to spherical symmetry, the ion flow *J* depends only on the distance *r* between the center of the system (sphere) and the selected point in space (Figure 3).

According to the law of mass conservation, the number of ions entering the spherical space with the radius *r* is equal to the number of ions adsorbed on the surface of the nucleus, therefore we can write:
(1)$$-4\pi r^{2}J\left(r\right)+4\pi {\left(r+dr\right)}^{2}*J\left(r+dr\right)=0$$

In the Equation (1) the formula *J(r + dr)* was developed into the Taylor series:(2)$$-4\pi r^{2}J\left(r\right)+4\pi (r^{2}+2rdr+{\left(dr\right)}^{2})*\left(J\left(r\right)+\frac{dJ}{dr}dr\right)=0$$

Ultimately, the differential Equation (2) takes the form:(3)$$\frac{d}{dr}\left(r^{2}J\right)=0$$

Integration of Equation (3) leads to the dependence of the ionic flux from the distance *r* from the center of the system:(4)$$J=\frac{C_{1}}{r^{2}},$$
where *C*_1_ is an integration constant. According to Fick’s first law, an ion flux can be written as:(5)$$J=-D\frac{dc}{dr},$$
where *D* is the diffusion coefficient, *c* is the concentration of calcium ions. Comparing Equations (4) and (5) and integrating we obtain a solution in the Equation (6):
(6)$$c\left(r\right)=\frac{C_{1}}{Dr}+C_{2}$$

The calculation of the integration constants *C*_1_ and *C*_2_ requires the assumption of boundary conditions. It was assumed that for *r* → ∞ the ion concentration is equal to *c*_0_, while on the grain surface with the radius *r* the concentration is zero.
$$c\left(\infty \right)=c_{0}$$
c(r)=0

The application of the above boundary conditions allows the determination of the integration constants *C*_1_ and *C*_2_ expressed by Equations (7) and (8).
(7)$$C_{1}=-c_{0}D*r$$
(8)$$C_{2}=c_{0}$$

Thus, the ionic flux on the nucleus surface with the radius *a*, can be expressed by the Equation (9):(9)$$J={\left|-D\frac{dc}{dr}\right|}_{r=a}=\frac{D*c_{0}}{a}$$

Thus, the total ion flux on the nucleus surface is:(10)$$I=4\pi aDc_{0}$$

Assuming that the grain density is equal to *ρ*, we can use the expression describing the mass increase of grain in time, which is equal to the total ion flux *I* on the nucleus surface, according to Equation (10). Therefore, the increase in the mass of grain over time is expressed by the Equation (11) as:(11)$$\frac{d}{dt}\left(\frac{4}{3}\pi a^{3}\rho \right)=4\pi a^{2}\rho \frac{da}{dt}=4\pi Da*c_{0}$$

Assuming that at the initial moment *t* = 0, the grain radius is equal to 0, the solution of the Equation (12) is:(12)$$t=\frac{a^{2}*\rho }{2D*c_{0}}$$

Based on the growth time, it is possible to determine the rate of grain growth by the Equation (13):(13)$$\frac{da}{dt}\sim \frac{1}{\sqrt{t}}\sim \frac{1}{a}$$

### 2.2. Data Used in Calculations

Based on the calculations, the analysis of the change in the grain radius over time was carried out. The following literature data were used in the calculations:

-Diffusion constant for calcium ions in muscle tissue: *D* = 1.65 × 10^−6^ (cm^2^/s) [34],

-The concentration of uncomplexed calcium: *c*_0_ = 42 mg/l = 4.2 × 10^−5^ (g/cm^3^), which is 50% of the total calcium in extracellular fluids [35].

The value of radius (*a*) and density (*ρ*) of grains come from experimental data from our previous work [30]. The density of CPPD and HAP grains with a radius of *a* = (0.02–0.1) cm in LF samples, extracted from the young patients and the elderly patients with stenosis, is presented in Table 1. The histogram of minerals density distribution in LF is presented in Figure 4.

## 3. Results

The examples of results of the model calculations, which show a dependence of the grain radius from the minerals growth time built with CPPD, are presented in Figure 5. The dependence of *a(t)* for HAP minerals in ligaments from the control group (35 and 36 y.o.) and the stenotic group (59 and 60 y.o.) are shown in Figure 6 and Figure 7, respectively. The differences between individual cases were insignificant, therefore the most diverse examples are presented. The maximum measurement error, shown in Figure 5, Figure 6 and Figure 7, was half of the voxel size. For the remaining cases, measurement errors did not differ significantly. The calculations assume that the values of *D* and *c*_0_ are constant for all cases.

In Figure 5 it can be seen that the growth time of CPPD grains ranged from ca. a month to 6 years, while in the case of HAP grains (Figure 6, Figure 7) it was up to 4 years for the young and up to 5.5 years for the elderly patients. The results of *da/dt* calculations for CPPD and HAP grains are shown in Table 2.

## 4. Discussion

A model of collagen mineralization was presented to estimate the growth time of mineral grains. The complexity of the system required a few simplifications. The most important was the assumption about the spherical shape of grains and the assumption that the growth environment is built only of calcium ions. Nevertheless, it is possible to develop the model with additional factors, as the following; a multi-component growth environment, competitiveness of ions between grains, partial dissolution of grains, fluctuations of internal environment conditions, adoption of plate or cylindrical shape, aggregation of nucleus, Ostwald ripening, etc. In the model calculation the constant diffusion *D* for the muscle tissue was used because of some similarities in the structure of tissues, however the difference between the muscle tissue and the ligament may be due to the viscosity of these tissues. In Figure 5 it can be seen that the growth time of minerals built of CPPD equals ca. a month to 6 years, while for HAP minerals it is up to 4 years for the young patients from the control group and up to 5.5 years for the elderly patients from the stenosis group. The value of *da/dt* decreases with the increase of the grain radius. It should be noted that the largest change in *da/dt* was noticed for deposits built of HAP. It suggests that CPPD deposition may differ from HAP and can be controlled by the factors involved in rheumatic and metabolic diseases, correlating with overproduction of inorganic pyrophosphate [7]. Factors affecting the rate of grain growth include diet, systemic diseases (diabetes, obesity), and used drugs that have a direct impact on the composition of physiological fluids and calcium metabolism. Some differences in *da/dt* values for HAP and CPPD may indicate different types of crystallite growth. At the same time, there are no significant *da/dt* differences between young and elderly patients, which suggests that the HAP mineralization in both groups of patients follows similar mechanisms. The analysis of the volume of the mineral at different time-lapses could provide more detailed information about the grain growth and the calcification mechanism. In the human body it is extremely difficult to observe the mechanism of mineral growth but an experiment carried out in vivo on the animal model or in vitro with the use of µ-CT measurements is feasible and awaits further investigation.

In the case of the knowledge of calcium density in the individual grains it would be possible to calculate the correlation between the calcium flux from the extracellular environment and an increase in grain mass in which the major contribution is calcium. However, the physicochemical analysis performed in our previous work [30] did not allow the determination of the calcium density in each grain. Thanks to the high resolution of µ-CT it was possible to calculate the density of the grains, built of HAP and CPPD. This approach would be considered in future models of calcification and requires the use of high-resolution methods of elemental analysis but needs further investigation.

The experimental data which describe the time of calcification growth in vivo are very sparing and relate to animal models. However, the results have been compared with models demonstrated in the literature [31,36]. The analysis was performed on mineral deposits culturing in vivo for 10 days. It was shown that the growth rate of mineral deposits with a diameter of 0.002 cm, extracted from rats, is about 2.1 × 10^−9^ cm/s [31], whereas for our model calculations the growth rate is equal to 1.75 × 10^−9^ cm/s for *a* = 0.02 cm. The significant difference in deposits size, which grow in a short time, unequivocally indicates a faster metabolic rate in animals compared to humans. Arroyo and colleagues [37] presented the calcification model in horse arterials. The mineral deposits with a size of 5 mm grew in about 4 years. Our results show that within 4 years, the minerals reach ~1.6 mm in diameter. The difference in the size of minerals may be due to the faster metabolic rate in horses as well as the localization of mineral growth. The rate growth of calcifications in arteries may be intensive because of the constant contact with the blood environment.

According to a computational model for aortic heart valve calcification, 25% of the leaflet was covered by calcification in about 6 years [36,38]. The computational model confirms our model calculations. In the same time of growth, a massive CPPD mineralization in LF was confirmed [30]. The analysis carried out on patients in [39] shows that calcifications with a volume of ~2.9 mm^3^ appears after 4.7 ± 0.8 years, which confirms the results of model calculations of mineral grains with a volume of ~3.04 mm^3^ growth for about 5 years.

Calcium metabolism disorder is most often diagnosed in parathyroid and kidney diseases, as well as in disorders of vitamin D metabolism [40]. The presence of chronic diseases and the use of drugs that affect calcium absorption, as well as individual differences and a change of eating habits, may cause a disturbance of the calcium balance. Changes in the ion concentration in physiological fluids may affect the size of grains and the growth time that have a promoting or inhibiting effect (Mg^2+^, Sr^2+^) on the mineralization process [41,42]. It is worth mentioning the inhibitors which play a major role in preventing ectopic mineralization. In vitro experiments show that osteocalcin (bone–Gla protein), osteonectin, and an elevated rate of osteopontin, induced by hyperphosphatemia, may have an inhibitory effect on matrix mineralization by binding to lattice calcium exposed at the crystal surfaces, regulating crystal dimensions [43]. The proteoglycans (i.e., decorin, lumican, biglycan) have important roles in matrix mineralization and may block active nucleating sites on the HAP surface slowing down the growth process [44,45].

Analyzing the decrease in *da/dt* values of grains with age, several factors that inhibit growth can be mentioned. First of all, the fastest growth can take place up to the saturation of all free chemical bonds in the micro-damage of the collagen matrix. Then, the grain grows outside the micro-damage. If the concentration of ions involved in mineralization remains constant, the invariable number of ions is adsorbed on the growing surface of the spherical grain, and thus the increase in grain radius takes place in a longer and longer time. The grain growth can also be decreased or inhibited by the change of diet or elimination of calcium interfering drugs on the influence on Ca absorption [46].

The second issue may be the partial dissolution of grains for the growth of others, which may also dissolve over time. The difference in the growth time of the CPPD and HAP grains suggests that the mechanism of mineralization does not follow identical pathways. A reflection of this phenomenon is that there are also more than five times a higher number of CPPD grains per 100 mm^3^ of tissue than in the case of HAP grains [23]. Differences in the number and size of grains suggest that they were in a different phase of growth, and the nucleation of grains probably occurred at different times. It should be mention that the mineral deposits had a heterogeneous structure. Hydroxyapatite was confirmed as the main mineral phase. However, chemical analysis confirmed the presence of a small amount of hydroxyapatite precursors (octocalcium phosphate and amorphous calcium phosphate) in a few cases [30]. The presence of the precursor structures can be included in the calculations for a detailed analysis of the growth mechanism.

In the model calculations, the concentration of free calcium ions was used. It should be noted that in extracellular fluid about 40% of calcium is bound to proteins, mainly albumin, while ~10% Ca is in the form of complexes with organic anions such as citrate and lactate anions. The remaining 50% is ionized calcium with high physiological significance. The assumptions adopted in the work do not take into account fluctuations in calcium concentration, and the results show the time of the grain growth with undisturbed calcium management.

The presented model is the first step in estimating the growth time of mineral grains in human soft tissues. It may be useful in experiments and in analysis of the calcification mechanism. This knowledge can be crucial in the elimination or reduction of harmful factors leading to loss of tissue elasticity not only in the spine ligaments case but also in stenosis of the aortic valve and cartilage and tendon mineralization. The growth rate may give information about the presence of additional factors promoting the growth of calcifications. The knowledge about the growth time of minerals in soft tissues may aid in estimating a relationship between the incidence in the human body (i.e., metabolism disease, stenosis of tissue or mechanical trauma related to sport) and the appearance of calcifications, as well as supplying information on the mineralization of breast, prostate, and thyroids, where the appearance of mineral grains is very often the first symptom of the tumor process [47,48,49,50].

## 5. Conclusions

The model results indicate that the growth time of mineral grains with a radius of up to 400 μm is relatively short, less than a year, and very difficult or impossible to identify by medical imaging techniques, such as magnetic resonance imaging and computed tomography. Larger grains with a radius of 1000 μm, visible in medical imaging methods, take about 6 years. Therefore, this pathology is only diagnosed at a higher stage as co-accompanying, usually on the occasion of diagnosis of other diseases. Some differences in *da/dt* values for HAP and CPPD suggest different types of crystallite growth. It is likely that additional factors are involved in the mechanism of CPPD growth. At the same time, there were no significant *da/dt* differences between the control and stenosis groups, suggesting that the HAP mineralization in both groups of patients follows similar mechanisms. It indicates that the spinal canal stenosis does not influence the calcification growth rate. Differences in the growth time of grains suggest that they were at a different stage of growth, and probably nucleation occurred at other times The proposed model can be developed with the use of additional factors, for example other ions in the extracellular fluid or fluctuation in ion concentration. Determination of the time of calcification growth in the yellow ligaments as well as in the case of other soft tissues has an important role in supplying information on the calcification mechanism and the relation between disease incidents and the appearance of pathological mineralization. It can support the elimination or reduction of the harmful factors that promote mineralization.

## Figures and Tables

**Figure 1 materials-13-02130-f001:**
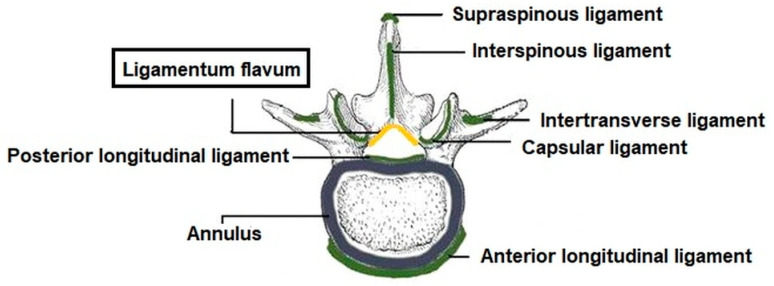
Ligament system stabilizing the spine.

**Figure 2 materials-13-02130-f002:**
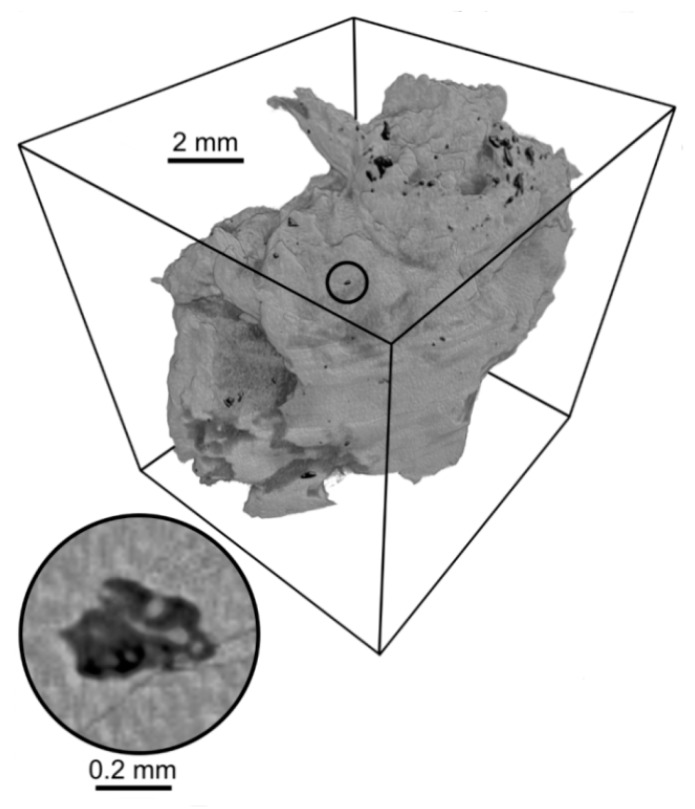
The reconstructed µ-CT image of the yellow ligament (stenotic group) with two threshold values. The upper threshold value (gray) refers to the soft tissue, the lower value (black) is mineral grain. Measurement with a resolution of 13.5 μm. The area measured with a resolution of 2 μm is marked. A two-dimensional cross-section of this area with the thresholding described above is shown [30].

**Figure 3 materials-13-02130-f003:**
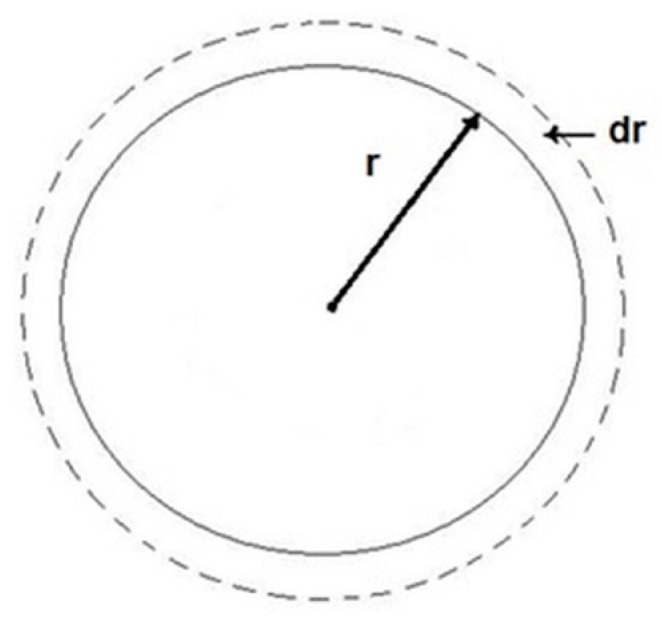
Schematic representation of grain growth with radius *r.*

**Figure 4 materials-13-02130-f004:**
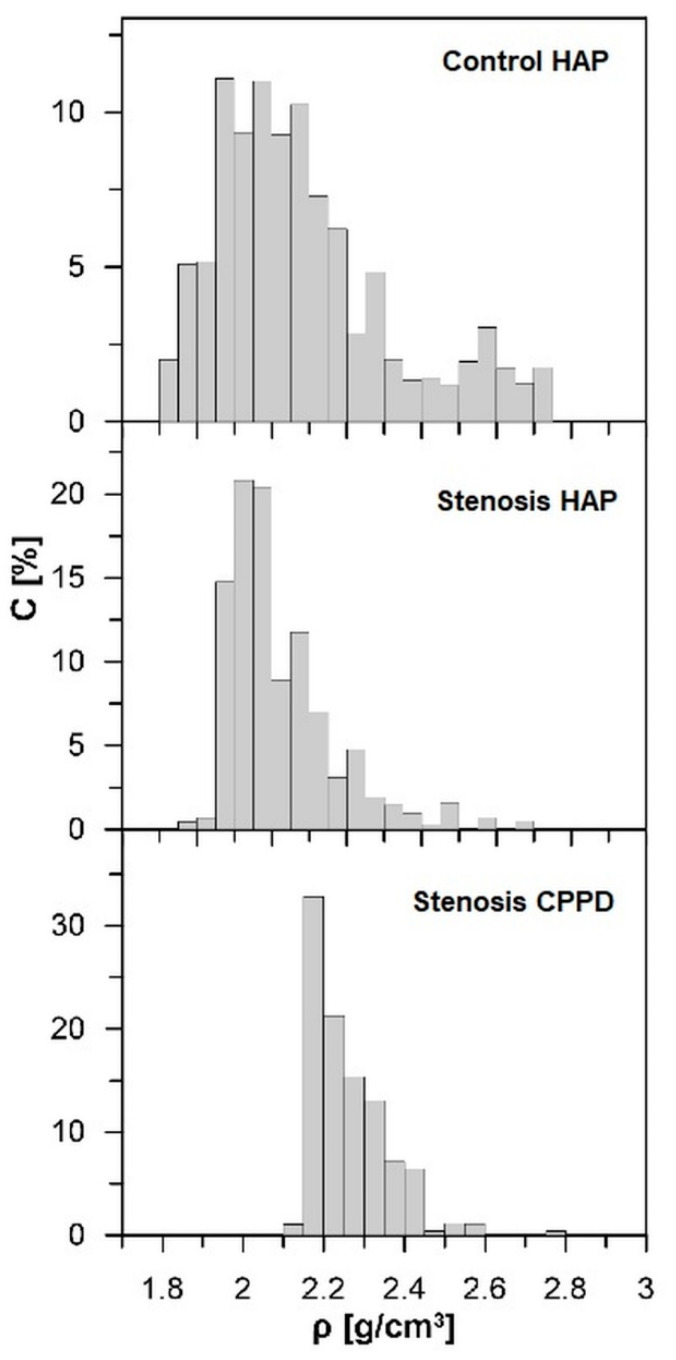
Histogram of grains density distribution (ρ) in ligamentum flavum samples. C—percentage contribution of minerals with given density in 100 mm^3^ of a sample [30].

**Figure 5 materials-13-02130-f005:**
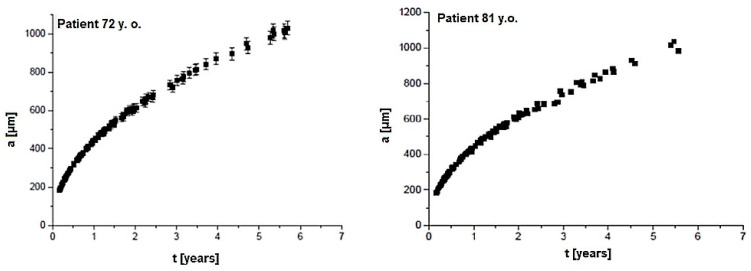
The dependence of the radius value of mineral grains built from calcium pyrophosphate dehydrate (CPPD) from the growth time for two cases of yellow ligaments.

**Figure 6 materials-13-02130-f006:**
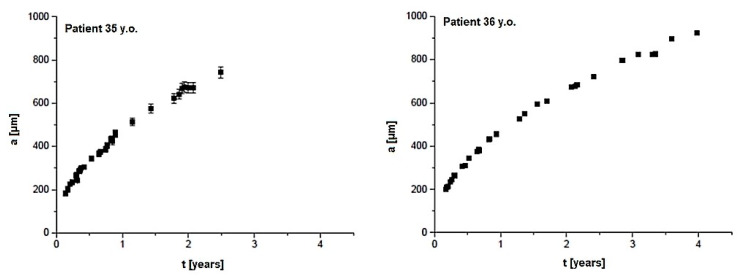
The dependence of the radius value of mineral grains built from HAP from the growth time for two cases of the yellow ligaments young patients (control group)**.**

**Figure 7 materials-13-02130-f007:**
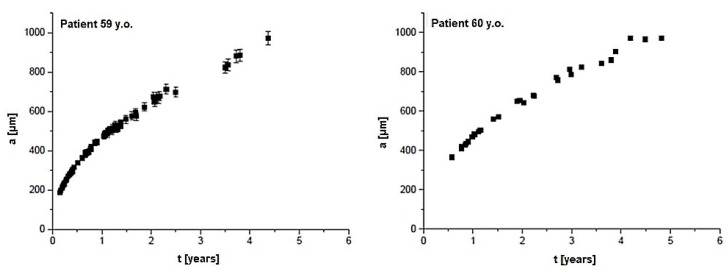
The dependence of the radius value of mineral grains built from hydroxyapatite-like mineral (HAP) from the growth time for two cases of the yellow ligaments of elderly patients (stenosis group)**.**

**Table 1 materials-13-02130-t001:** The range of density (*ρ)* of calcium pyrophosphate dehydrate (CPPD) hydroxyapatite-like mineral (HAP) grains with a radius of *a* = (0.02–0.1) cm from control and stenosis groups of patients.

	Age (Years)	Density (g/cm^3^)
CPPD	72	2.15–2.25
CPPD	81	2.61–2.40
HAP control	35	1.78–1.95
HAP control	36	1.85–2.0
HAP stenosis	59	1.85–2.0
HAP stenosis	60	1.87–2.20

**Table 2 materials-13-02130-t002:** *da/dt* for the minimum and maximum radius (*a*) of CPPD and HAP grains for selected cases of age.

	Age (Years)	A = 0.02 cm	A = 0.1 cm
CPPD	72	1.75 × 10^−9^ cm/s	2.9 × 10^−10^ cm/s
CPPD	81	1.54 × 10^−9^ cm/s	3.0 × 10^−10^ cm/s
HAP-control	35	1.89 × 10^−9^ cm/s	4.78 × 10^−10^ cm/s
HAP-control	36	1.78 × 10^−9^ cm/s	3.7 × 10^−10^ cm/s
HAP-stenosis	59	1.76 × 10^−9^ cm/s	3.56 × 10^−10^ cm/s
HAP-stenosis	60	1.0 × 10^−9^ cm/s	3.2 × 10^−10^ cm/s

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
