# Peer review of "Model of Pathological Collagen Mineralization Based on Spine Ligament Calcification"

_materials, 2020, doi:10.3390/ma13092130_

Round 1
Reviewer 1 Report
The manuscript does not show major shortcomings but at times becomes difficult to read and uses an obscure terminology. For instance, I am not sure of what the authors, speaking of mineral nucleation, intend for "structural defects" in "collagen fibres". Usually the word "fibrils" is used to designate the typical collagen aggregates that show the 67nm cross-banding and are only visible with electron microscopy, while "fibres" indicates bundles of fibrils, large enough to be visualized by histological techniques (light microscopy).
It is commonly agreed that nucleation begins in the gap zone of collagen fibrils: is this is what the authors mean, then "fibre" is the wrong term and the gap zone is an intrinsic feature of the molecular stagger and not a "micro-damage". On the other hand, the relatively large size of the mineral particles seem hardly attributable to the ultrastructural features of collagen aggregates.
Im my opinion, although the experimental part seems reasonably good, the manuscript needs an extensive revision before it can be considered for publication.
Author Response
We are very grateful to the Reviewer for valuable comments that help to improve this work making it more legible and useful emphasizing the information which can be a step to better understanding the calcification process and can provide complementary information for future experiments. The detailed description of the corrections is provided -please see the attachment.

Reviewer 2 Report
General comments:
This study presents a mathematical model predicting the growth rate of mineral phase in soft tissue based on their experimental data obtained from human samples. The spherical symmetry model used in this study is straight forward to understand although they used several assumptions which too simplifies the complex biological system. While it will be meaningful to report the analyses of micro CT data obtained from human samples of different groups, the manuscript needs to be improved significantly for the publication in Materials in several ways. Please see the commented below.
Major comments:
- The term “collagen structure mineralization” in the title is somewhat unclear to me. Indeed, the model used in this study does not reflect any physicochemical property of the collagen structure.
- The mechanisms of mineralization of bones and soft tissues used in this study would be different. I have a few comments for the 1st paragraph of the introduction as below.
- In lines 34-37, “Collagen fibers occurring in bones demonstrate specific structural defects endowed with an electric charge where the nucleation of bone apatite can be started. Collagen is not a crystallization promoter but the process is assisted by matrix vesicles released from the osteoblast cell membrane.” I do not agree with these two sentences if authors are describing about compact bone mineralization process. It would be better if you can define the meaning of “defects”. Do collagen fibers in bones naturally have certain portion of defects (compact bones are highly organized)? Are they are mineralized only when there are defects? If so, why all the bones are mineralized? Several recent articles emphasize active role of collagen to promote apatite crystallization. Authors may underestimate this aspect.
- In following sentences, the authors’ description about the “structural defects” is also confusing. Do authors indicates the gap zone (~40 nm) of collagen fibrils caused by staggered assembly of fibrils. Also, the expression in lines 39-40 (“collagen fibers and structural defects aggregate with each other along the long axis.”) may not be correct. Perhaps, defects may form during the aggregation of collagen fibrils forming a long fiber, and defects may form in this process, instead of fibers and defects aggregate each other. It is not reasonable to me that “fibers and defects aggregate each other”.
- 5-7 and Table 1 are less convincing. Do authors calculate t from a values reported previously? How do you prove that the grain size is a function of the time hypothesized in your model development (as shown in equation 13)? Can it be a function of defect size? Do you have any experimental data collected at different time lapses? What are error bars?
- Therefore, the importance of determining the growth time in this study is weakly presented. Authors stated that the modeling is important for “preventing loss of tissue elasticity”, in the last paragraph of the discussion. It seems like the row data, such as the total content of mineral phases or size distribution can also be used for prediction. What additional information can the modeling results of growth time provide?
- In discussion, please comment the possible difference between diffusion of calcium ions in muscle tissues and yellow ligaments because authors used diffusion constant for calcium ions (in line 149).
Minor comments:
- In the abstract, authors used mixed tenses. In general, the abstract need to be revised. For example, growth rate or growth kinetics would be more appropriate than growth time.
- In line 34, the mineral phase of bone is different from hydroxyapatite in several ways. Authors need to replace the term “hydroxyapatite” to “hydroxyapatite-like mineral”, or “bone apatite” as used in elsewhere of the manuscript.
- In line 68, replace ‘not typical’ to ‘non typical’.
Author Response
We are very grateful to the Reviewer for valuable and detailed comments that help to improve this work making it more legible and useful emphasizing the information which can be a step to better understanding the calcification process and can provide complementary information for future experiments. The detailed description of the corrections is provided- please see the attachment.
Dr. Sylwia Orzechowska
Prof. Renata Świsłocka
Prof. Włodzimierz Lewandowski

Reviewer 3 Report
In the presented work, the authors aim to model growth of mineral particles in spine ligaments using a diffusion model that focuses on the transport of Ca ions through the soft tissue and subsequent mineral growth. Thus, the authors obtain a relationship between mineral particle size and duration since nucleation. Eventually, size and density information of such mineral particles from their former work [ref 23] are used and plotted in the context of the derived relationship.
Growth of mineral particles in cases of ectopic/pathological mineralization, as it happens in soft tissues due to stress induced micro damage, is a highly interesting topic and the in-vivo kinetics of the related processes are difficult to assess experimentally due to limitations of sample preparation. Hence, mathematical or computational models of particle growth are in general highly appreciated as they can provide complementary information to experiments thus justifying approaches like the one presented in the manuscript.
Unfortunately, the presented work exhibits fundamental problems in the introduced model, unclear statements and consequently over-interpretation of the obtained results, so that I cannot recommend this work for publication. Details are listed below. In general the style of writing makes the manuscript quite hard to read. Often sentences are out of the context or unclear what they refer to, and statements lack proper references. Expectations raised in the abstract are not fulfilled and the biochemical/biological background is very weak.
Major concerns:
Abstract:
The first sentence introduces the aim, the mathematical modelling of minerals in human ligaments correctly, but continuing with. “The biological material….” is out of the context here, also the type of samples is not mentioned. Further, µCT measurements are mentioned which were not conducted in this study but taken from literature data. Also the determination of the constituents of the mineral phase were not conducted in the presented study as suggested in the abstract. For the sentence “The growth time of CPPD grains ranged from….” it is unclear if this is a result of the experiments or the model. There is no solid basis for the conclusion (last sentence of the abstract).
Introduction:
- The authors emphasize in line 32/33 that there are similarities between pathological and physiological mineralization and introduce bone formation involving matrix vesicles (line 37) as physiological process. Taking this into account, it is surprising that the different mineralization concepts (matrix vesicles vs. passive diffusion) are not further discussed as the introduced model is based on latter only.
- The introduction hardly gives any motivation for the conducted study. Why is it important to gain knowledge about the growth kinetics of minerals in the ligament?
- The focus of the paper is a mathematical diffusion model to obtain information on the time scale of mineral growth. The introduction does not contain any literature/background about previous modelling approaches and how they differ to the presented one. Solely, the statement in line 84 mentions that other models where used for animal tissue without giving any reference. Instead, the introduction is mainly about medical background which is not really the topic investigated in the study.
Materials and Methods
- The whole first chapter refers to literature data obtained by the same group without explicitly mentioning this in this section. I don’t see the reason why FTIR, XRF, XRD and µCT are introduced in the material and methods section if they were not conducted in the presented study. If literature values are used as input parameters for the model it is sufficient to simply refer to the corresponding paper.
- Line 108, Line 17: “Ca is mentioned as the main component of HAP.“ This is a very problematic statement as the other elements of HAP cannot be neglected. In stoichiometric HAP ( Ca10(PO4)6HO2 ) the P and O cannot be neglected. The mass of Ca in the unit Cell of around 400u is exceeded by the mass of O (416u) and also P (186u) is not neglect able. This has a fundamental impact on the model where the Ca ion flux is directly correlated with the density of the mineral: line 140: “Assuming that the grain density is equal to ρ, we can use the expression describing the mass increase of grain in time, which is equal to the total ion flux I on the nucleus surface” this statement is simply not correct. Using then later on the material density derived from µCT data as density (g/cm3) without considering that HAP is not pure Ca makes all further considerations and comparisons very problematic. Similar considerations hold for Calciumpyrophosphat – maybe to even stronger extend due to the higher abundance of P.
- Line 166: the authors mention that differences between the individuals were insignificant. How was this evaluated. In the abstract the two groups “stenosis vs. control” are emphasized. This comparison is not present in the results and in general throughout the paper.
Discussion
- Line 219: The discussion regarding calcium metabolism disorders is problematic since mineralization inhibitors are not mentioned (proteoglycans, osteopontin pyrophosphate…) but play a major role in preventing ectopic mineralization.
- Line 227: As Ca is under careful homeostatic control it is highly unlikely that the diet has an influence on the free ion Ca level.
Conclusion
- Line 246: “Determination of the mineralization growth time has an important preventing role against the loss of tissue elasticity”. How can the determination have a preventive role?
- Line 248: “Model calculations indicate that the growth time of mineral grains with a radius of up to 400 μm is relatively short and very difficult or impossible to identify by medical imaging techniques.” These “medical imaging techniques” are only mentioned in the abstract and the conclusion and this topic is not discussed – it is unclear to which techniques the authors refer to.
- In general, non of the conclusions are supported by the results reported in this study. A couple of new topics are raised here which were not part of the discussion.
Minor comments:
Line 44: reference is missing
Line 45: I guess that “tissue calcification” here refers to the pathological mineralization of soft tissue.
Line 50: The sentence “The hypothesis concerning….” is out of context
Line 59/60: reference is missing
Line 61/62: reference is missing
Line 71: “The prevention of soft tissue calcification is a serious problem,…” I think the opposite is the case.
Line 84: reference is missing – I don’t believe that all animals have a faster metabolism compared to humans
Line 211/213: the statements are out of context and without reference
Author Response
We are very grateful to the Reviewer for valuable and detailed comments that help to improve this work making it more legible and useful emphasizing the information which can be a step to better understanding the calcification process and can provide complementary information for future experiments. The detailed description of the corrections is provided- please see the attachment.
Sylwia Orzechowska, PhD
Prof. Renata Świsłocka
Prof. Włodzimierz Lewandowski

Reviewer 4 Report
The authors analyzed in this study the growth time of minerals in human ligaments using the mathematical model. 15 samples (55-81 years old) and 9 samples 20-53 were used. The chemical structure of deposits was analyzed using FTIR, XRF and XRD techniques. Also, Samples were analyzed using computed microtomography where the volume, density and the number of mineral deposits has been calculated. The main conclusion is that the change of growth rate is the largest for HAP deposits.
1. It would be nice if the authors could mention Urry´s charge neutralization theory for calcification in the introduction section. (Urry DW. PNAS 1971; 68:810-814).
2. The analysis was carried out for two chemical structures of deposits- HAP and CPPD. Is it possible to analyze the two precursors of HAP (octocalcium phosphate and amorphous calcium phosphate)? What percentage of HAP precursors are in the samples? The authors could mention this HAP precursors in introduction or discussion section.
Author Response
We are very grateful to the Reviewer for valuable comments that help to improve this work making it more legible and useful emphasizing the information which can be a step to better understanding the calcification process and can provide complementary information for future experiments. The detailed description of the corrections is provided- please see the attachment.
Sylwia Orzechowska, PhD
Prof. Renata Świsłocka
Prof. Włodzimierz Lewandowski

Round 2
Reviewer 2 Report
In the revised version of the manuscript, authors well clarified the aspects that this reviewer was uncertain. As authors stated, it would be meaningful to estimate the time of the mineral growth from a human data at a specific time. However, I still think that the analysis of the data at different stages from a same person can prove their hypothesis. I don't strongly urge this, but authors may consider adding this aspect as possible future work in the manuscript.
Author Response
Model of pathological collagen mineralization based on spine ligaments calcification, ID: materials-765847
Response letter to the Reviewer
We very much appreciate the comments, cooperation, and valuable suggestions of the Reviewer aiming at clarifying any doubts. According with the Reviewer suggestion, we have added the additional information in the text, line: 249-253.
Sincerely Yours,
Sylwia Orzechowska, PhD
Prof. Renata Świsłocka
Prof. Włodzimierz Lewandowski
Reviewer 3 Report
I appreciate the effort that the authors put into the present manuscript which clearly improved. However, one major concern was is in my opinion still not addressed enough, namely the focus on Ca ion diffusion for the mathematical model. In the response letter the authors state:
“The XRF analysis of mineral deposits showed that the concentration of Ca is about 17%, P (~7.3%) and Ca/P ratio equals to 1.8±0.8. The elemental analysis demontrated also the presence of other elements. However, the Ca concentration is the highest. It is a reason why we consider Ca as a main component of mineral phase in deposits.”
Here I still see the problem that:
- The chosen method for elemental analysis in reference [30] does not consider light elements like Oxygen. As HAP hast around 26 O atoms and only 10 Ca even in wheigtpercent the fraction of O is expected to exceed the fraction of Ca hence the term “main component” is still problematic.
- As the authors state in the response letter that the concentration of Ca in the investigated minerals is about 17% and consequently that 83 % of the elements are not Ca.
However, the model implies that the total Ca ion flux onto the particle surface (Formula 10) equals the increase of mass density in g/cm^3 (Formula 11). How can this be understood if Ca makes up only 17% of the elements present in the particle and hence 83% of the elements are the rest. I totally agree that models start with rough estimations and simplifications – but this seems not appropriate to me. Wouldn’t it be better to roughly estimate the Ca density in the particle based on the previously determined composition and correlate this value with the Ca flux at the surface?
Author Response
We very much appreciate the comments and valuable suggestions of the Reviewer aiming at clarifying any doubts. According to the Reviewer suggestion, we have added the additional information. Please see the attachment.
Sincerely Yours,
Sylwia Orzechowska, PhD
Prof. Renata Świsłocka
Prof. Włodzimierz Lewandowski

Round 3
Reviewer 3 Report
The the paragraph added by the authors adds some additional information for the reader about ways to further develop the introduced model. From my point of view, this study still describes only basic a starting point for developing more realistic models. As this is now sufficiently mentioned in the latest manuscript i have no objections to publishing the manuscript in the present from.